# Multimodal cell tracking from systemic administration to tumour growth by combining gold nanorods and reporter genes

Joan Comenge[1‡], Jack Sharkey[2,3], Oihane Fragueiro[4], Bettina Wilm[2,3], Mathias Brust[4], Patricia Murray[2,3], Raphael Levy[1†*], Antonius Plagge[2,3†*]

[1]Institute of Integrative Biology, University of Liverpool, Liverpool, United Kingdom; [2]Department of Cellular and Molecular Physiology, University of Liverpool, Liverpool, United Kingdom; [3]Centre for Preclinical Imaging, Institute of Translational Medicine, University of Liverpool, Liverpool, United Kingdom; [4]Department of Chemistry, University of Liverpool, Liverpool, United Kingdom

*For correspondence:
rapha@liverpool.ac.uk (RL);
a.plagge@liv.ac.uk (AP)

†These authors contributed equally to this work

Present address:
‡Nanotargeting S.L, Barcelona, Spain

**Abstract** Understanding the fate of exogenous cells after implantation is important for clinical applications. Preclinical studies allow imaging of cell location and survival. Labelling with nanoparticles enables high sensitivity detection, but cell division and cell death cause signal dilution and false positives. By contrast, genetic reporter signals are amplified by cell division. Here, we characterise lentivirus-based bi-cistronic reporter gene vectors and silica-coated gold nanorods (GNRs) as synergistic tools for cell labelling and tracking. Co-expression of the bioluminescence reporter luciferase and the optoacoustic reporter near-infrared fluorescent protein iRFP720 enabled cell tracking over time in mice. Multispectral optoacoustic tomography (MSOT) showed immediate biodistribution of GNR-labelled cells after intracardiac injection and successive clearance of GNRs (day 1–15) with high resolution, while optoacoustic iRFP720 detection indicated tumour growth (day 10–40). This multimodal cell tracking approach could be applied widely for cancer and regenerative medicine research to monitor short- and long-term biodistribution, tumour formation and metastasis.
DOI: https://doi.org/10.7554/eLife.33140.001

## Introduction

Non-invasive optical imaging methods for preclinical *in vivo* research include bioluminescence (BLI) and fluorescence as well as photoacoustic/optoacoustic tomography, a technology that has only been developed recently (*Deliolanis et al., 2014*; *Wang and Yao, 2016*; *Weber et al., 2016*). These imaging modalities have enabled great progress in the tracking of labelled cells longitudinally over time in animal models of disease, which has become especially relevant for cancer research and cell-based regenerative medicine therapies (*de Almeida et al., 2011*; *James and Gambhir, 2012*; *Sharkey et al., 2016*). The resolution and sensitivity of optical imaging in animals is limited by auto-fluorescence, absorption and scattering of excitation and/or emission light, especially in deep tissues. The optimal window for *in vivo* optical imaging lies in the near infrared (NIR) spectrum (~650–900 nm), since absorption through the main endogenous chromophores (oxy-haemoglobin, deoxy-haemoglobin, melanin, water and lipids) are minimal in this spectral range (*Weber et al., 2016*). For permanent cell labelling and tracking, genetic modification with reporter genes is the method of choice, although fluorescent tags and nanoparticles have been developed recently for sensitive

**eLife digest** Many scientists are studying the possibility of using human cells to treat diseases. For example, using stem cells to regenerate damaged body parts or genetically engineered immune cells to destroy cancer. Scientists need new tools to track what happens to these cells once they have been injected into a laboratory animal. This will help them understand how they work and make sure these potential treatments are safe. One concern with using cells as a treatment is that they might form cancerous tumors.

To track these cells in a laboratory animal, scientists need two things: a way to distinguish the treatment cells from the animal's normal cells and an imaging tool that allows them to see where the cells are in a living animal. One way to differentiate treatment cells from normal cells is to genetically engineer them to make a fluorescent protein called iRFP720. Another way is to fill the cells with gold nanorods. Both, the fluorescent protein and the gold nanorods, absorb light in the infrared range. Scientists can use a technique called multispectral optoacoustic tomography, which transforms infrared light into ultrasound signals to create an image, to see where these markers are in the body.

Now, Comenge et al. showed that the gold nanorods and multispectral optoacoustic tomography track the cells immediately after injection into the blood stream of a mouse. Most of the injected cells die within a few days, and the nanorods are progressively eliminated from the body through the liver. But some of the injected cells live on, multiply, and form tumors within a month. This was expected because the cells they used were chosen for their ability to sometimes form tumors. Using multispectral optoacoustic tomography to track the cells making iRFP720, Comenge et al. were able to see exactly where the tumors are deep inside the body.

Together, gold nanorods and iRFP720 could allow scientists to track where the cell-based therapies for cancer or other diseases go in the short and long term. This may help them prove whether these treatments work, and whether they have harmful effects. Comenge et al. are helping other scientists to use these techniques by distributing their tool for making iRFP720-producing cells.

DOI: https://doi.org/10.7554/eLife.33140.002

short-term cell tracking over a period of a few cell divisions (*Comenge et al., 2016*; *Dixon et al., 2016*).

Using luciferase reporter genes, bioluminescence constitutes the most sensitive optical modality due to its excellent signal-to-noise ratio, as light emission only occurs in the presence of a functional enzyme and its required co-factors. Firefly, luciferase has become the most widely used reporter as its substrates, D-luciferin or CycLuc1 (*Evans et al., 2014*), are very well tolerated by animals and, compared to other luciferases, its peak light emission at around 562 nm is closest to the infrared window for in vivo imaging (*de Almeida et al., 2011*). Although highly sensitive *in vivo* cell tracking via bioluminescence imaging of firefly luciferase is well established (*de Almeida et al., 2011*; *Mezzanotte et al., 2013*), this modality provides poor information about the spatial localisation of cells. Fluorescence has recently gained importance for animal imaging, since novel near-infrared fluorescent proteins (iRFPs) were developed from bacterial phytochrome photoreceptors (*Shcherbakova et al., 2015*; *Shcherbakova and Verkhusha, 2013*). Similar to bioluminescence imaging, fluorescence only allows limited spatial resolution due to the high scattering coefficient of photons in tissues.

On the other hand, photoacoustic imaging is based on the generation of ultrasound waves after absorption of light emitted by a pulsed laser. The sound waves are well transmitted in fluid media and less prone to scattering through tissues than emitted light. In fact, acoustic scattering is three orders of magnitude less than photon scattering (*Wang and Hu, 2012*), which overcomes deep tissue spatial resolution drawbacks of other optical-based imaging technologies. Interestingly, some iRFPs, such as iRFP720, have an absorption profile in the NIR window, thus enabling their use as reporter genes for photoacoustic imaging, and allowing deep tissue imaging and tumour monitoring in mice (*Deliolanis et al., 2014*; *Jiguet-Jiglaire et al., 2014*). For example, new iRFPs have been proven to be useful genetic photoacoustic reporters in mammary gland and brain tumour monitoring, which establishes them as dual-modality imaging probes (*Deliolanis et al., 2014*; *Filonov et al.,*

*2012*; *Krumholz et al., 2014*). In addition, in multispectral optoacoustic tomography (MSOT), a rapid multiwavelength excitation allows the distinction between different absorbers simultaneously after applying multispectral unmixing algorithms (*Tzoumas et al., 2014*). Hence, a number of endogenous (e.g. deoxy- and oxyhaemoglobin) or exogenously introduced targets can be imaged *in vivo* via MSOT, including fluorescent proteins and nanoparticles.

Since light absorption triggers the photoacoustic response, contrast agents with a high molar extinction coefficient facilitate their distinction from endogenous chromophores (*Lemaster and Jokerst, 2017*). In this regard, GNRs are an attractive choice since the extinction coefficients are in the range of 4–5.5 $\times$ $10^9$ $M^{-1}cm^{-1}$ for longitudinal surface plasmon between 728 and 845 nm (*Orendorff and Murphy, 2006*), that is orders of magnitude higher than endogenous absorbers or other types of contrast agents (e.g. AlexaFluor750 and iRFP720 coefficients are $2.5 \times 10^5$ and $9.6 \times 10^4$ $M^{-1}cm^{-1}$, respectively [*Shcherbakova and Verkhusha, 2013*; *Ntziachristos and Razansky, 2010*]). We recently demonstrated how, after coating GNRs with a 35 nm silica shell, the optical properties of GNRs are maintained even in harsh environments such as the endosomal vesicles (*Comenge et al., 2016*). This enabled very sensitive photoacoustic imaging of labelled cells, pushing the limits of detection into a range useful for biologically-relevant applications including cell tracking (*Wang and Jokerst, 2016*). However, cell division and cell death result in a decrease of GNR concentration and therefore a decrease of photoacoustic intensity. Thus, for long-term photoacoustic tracking of cells, adequate reporter genes are required.

Here, we characterise newly generated lentivirus-based bi-cistronic reporter gene vectors and silica-coated GNRs as synergistic tools for co-labelling of cells and tracking in multiple imaging modalities at different stages after cell administration. The GNR-860 offered an excellent photoacoustic sensitivity that enabled detection of a very low number of cells. Reporter gene vectors mediated co-expression of the most sensitive bioluminescent reporter, firefly luciferase, and the most red-shifted near-infrared fluorescent protein, iRFP720, which also acted as a photoacoustic reporter in MSOT. Whilst GNR-860 enabled early detection of cells by photoacoustic tomography after systemic injection, reporter genes were needed to monitor any potential tumour growth with the same technique. In addition, co-expression of luciferase allowed bioluminescence imaging. Although this modality does not offer the spatial resolution of photoacoustic tomography, it was used here to continuously monitor cell viability as well as for validating the photoacoustic results. The combination of the two cell labelling strategies (nanoparticles and genetic labelling) allowed short-term and long-term multimodal imaging of deep tissues with high spatial resolution without compromising sensitivity. It enabled the non-invasive monitoring of the initial cell biodistribution on a sub-organ level as well as the detection of tumour formation from very early stages, thus opening opportunities for gaining a better understanding of the safety of stem cell therapies, which is required for moving towards clinical translation (*Ankrum and Karp, 2010*).

## Results and discussion

### Gold nanorod synthesis and cell labelling

Two types of GNRs with distinct optical spectra were prepared following previously published protocols (*Nikoobakht and El-Sayed, 2003*; *Ye et al., 2012*). The rationale for evaluating two types of gold nanorods in this study is as follows. First, detection of GNRs at different wavelengths would support the versatility of the approach, for example when combining them with other reporter molecules, that is it is not dependent upon preparing the exact same nanorods as reported here. Second, having two different probes provided us with a powerful control for false positive signals as explained in more detail below in the context of Figure 4. The first batch consisted of GNRs with a longitudinal surface plasmon resonance (LSPR) peak at 710 nm (GNR-710). Their core size was 60.0 ± 7.5 nm by 25.7 ± 3.7 nm with an average aspect ratio of 2.3 ± 0.3. The second batch had a LSPR peak at 860 nm (GNR-860). In this case, their size was 93.8 ± 9.3 nm by 25.2 ± 4.4 nm with an average aspect ratio of 3.8 ± 0.5) (*Figure 1a*). Both GNRs were prepared to have LSPR bands in the NIR to favour their detection deep inside tissues.

We have recently demonstrated that a 35 nm silica shell is needed to minimise plasmon coupling between GNRs after cell uptake (*Comenge et al., 2016*). This enables the preservation of intrinsic optical properties resulting in an increased sensitivity for photoacoustic detection. Hence, both GNR

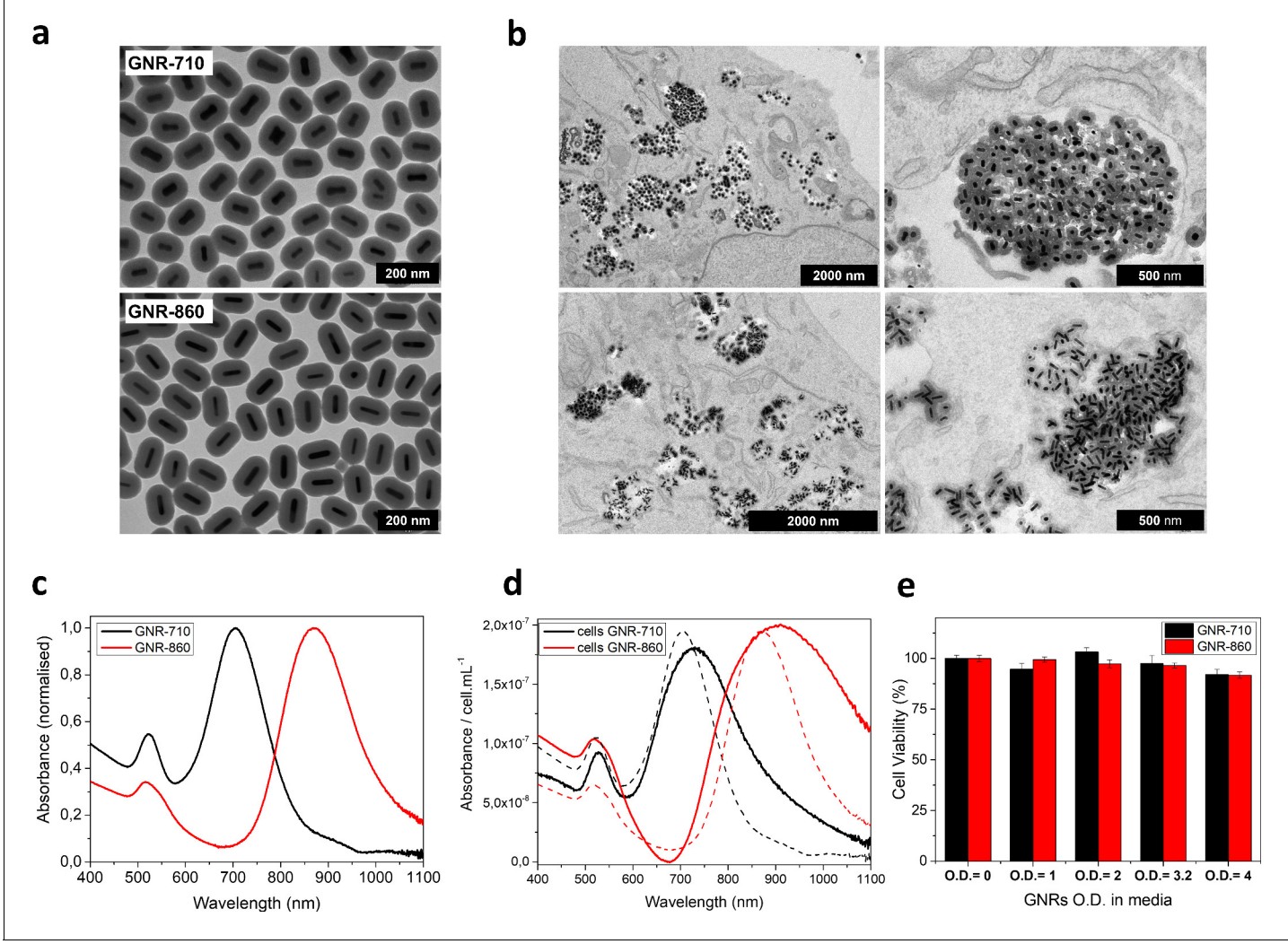

**Figure 1.** GNR characterization and cell labelling. (a) Representative transmission electron microscopy (TEM) pictures of GNR-710 (top) and GNR-860 (bottom). (b) TEM pictures of GNR-710 (top panels) and GNR-860 (bottom panels) inside cells. In both cases, the silica shell provides steric hindrance to minimise plasmon coupling in cellular vesicles. (c) Vis-NIR spectrum of GNRs in solution. (d) Vis-NIR spectrum of cells labelled with the corresponding GNRs. Dashed lines correspond to GNRs in solution. (e) Cell viability at different optical densities. Viability was assessed in triplicate and results are given as %±SD relative to cells that were cultured without GNRs.

DOI: https://doi.org/10.7554/eLife.33140.003

The following figure supplement is available for figure 1:

**Figure supplement 1.** Additional images of GNR-710 and GNR-860 uptake by MSCs.

DOI: https://doi.org/10.7554/eLife.33140.004

batches were coated with a silica shell as previously described (*Comenge et al., 2016*). Specifically, GNR-710 and GNRs-860 nm have a silica shell thickness of 38.7 ± 1.9 nm and 35.7 ± 1.6 nm, respectively (*Figure 1a*).

A murine MSC line was selected for labelling and tracking, since these cells have the capability to differentiate into mesenchymal derivatives, but can also form tumours occasionally (*Comenge et al., 2016*; *Kuzma-Kuzniarska et al., 2012*). The cells, which had first been transfected with a reporter gene vector (see below), were incubated with different concentrations of GNR-710 and GNR-860 for 24 hr. To provide a better comparison between GNRs of different sizes, we standardize and report the GNR concentrations by the optical density (O.D.) of the peak, because absorption is what triggers the photoacoustic response. For reference, O.D. = 4 corresponds to ~100 pM GNR-710

and ~72.5 pM GNR-860 (calculated by relating the absorbance at 400 nm of GNRs before silica coating to concentrations of molecular gold) (*Pastoriza-Santos and Liz-Marzán, 2013*).

To characterize their uptake, cells were incubated with GNRs at O.D. = 3 for 24 hr. TEM images of cells show GNRs localised inside vesicles, with silica shells separating the gold cores (*Figure 1b*, and *Figure 1—figure supplement 1*). In line with our previous report, the optical properties of GNRs were largely preserved after cell uptake (*Figure 1c,d*). Preservation of the absorbance intensity and the shape of the plasmon band are prerequisite for an optimal photoacoustic detection of GNR-labelled cells since it relies on a spectral deconvolution to differentiate intrinsic absorbers from probes.

To confirm that the cell labelling does not cause any toxicity, cell viability after exposure to a range of GNR concentrations up to O.D. = 4 was assessed (*Figure 1e*). Only at the highest GNR concentration was a slight decrease in cell viability observed: 92.0 ± 2.6% and 91.7 ± 1.6% for GNR-710 and GNR-860, respectively. These results are in agreement with our previous work, in which we showed that silica-coated GNRs did not affect cell viability, proliferation or differentiation potential of the mMSCs (*Comenge et al., 2016*).

## Generation of iRFP720-Luciferase bi-cistronic lentivirus reporter gene vectors and characterisation of labelled cells

To combine the sensitivity of bioluminescence imaging with the spatial resolution of MSOT, we generated bi-cistronic lentivirus vectors for expression of firefly luciferase and iRFP720 from the general eukaryotic EF1α promoter (*Figure 2a*). We compared vectors containing either an internal ribosome entry site (IRES) (*Hellen and Sarnow, 2001*) or a self-cleaving 2A element (*Szymczak and Vignali, 2005*; *Szymczak et al., 2004*), to determine the most efficient method of translating the second open reading frame (ORF; encoding luciferase) from a single mRNA. Following transfection with lentivirus vectors, the mMSCs were selected by FACS for iRFP720 fluorescence, to obtain a pure IRES-vector and a pure E2A-vector transfected cell population, respectively, with similar levels of expression of the first ORF protein (iRFP720) (*Figure 2b*). Co-expression of iRFP720 and luciferase in individual cells was confirmed by immunocytochemistry (*Figure 2c*), which also indicated that, in contrast to luciferase, iRFP720 was preferentially localised in the nucleus. To compare the levels of expression of luciferase from the IRES-vector and E2A-vector transfected cells, fluorescent and bioluminescent signals were quantified in cell suspensions using an IVIS Spectrum imager. For both types of reporter cells (IRES-vector and E2A-vector transfected), the fluorescence signal intensity for iRFP720 was the same, but the E2A-vector-transfected cells showed a five times higher luciferase bioluminescence (*Figure 2d,e*). These data indicate that in the context of our bi-cistronic lentivirus vectors, a superior translation efficiency of the second ORF from a single mRNA was achieved via an E2A element as compared to an IRES element. We therefore used the pHIV-iRFP720-E2A-Luc vector in all further experiments described herein. Proof of principle of the application of this vector for multi-modal imaging was tested in preliminary experiments, in which cells were injected subcutaneously and monitored with bioluminescence, fluorescence and photoacoustic imaging (*Figure 2—figure supplement 1*).

## Imaging biodistribution of cells after intracardiac injection

Detection of labelled cells after subcutaneous injection has provided indications that gold nanorods (our previous work [*Comenge et al., 2016*]) and genetic reporters (*Figure 2—figure supplement 1*) could enable cell tracking. Similar proof of principle of the utility of nanoparticles as contrast agents for photoacoustic cell detection has often been obtained by imaging labelled cells implanted locally (*Jokerst et al., 2012*; *Nam et al., 2012*). Here, we address the much more challenging case of a systemic injection where cells spread through the body and the signal, therefore, becomes diluted. Specifically, $1.5 \times 10^6$ mMSCs transduced with the iRFP720-E2A-Luc lentivirus and labelled with GNRs were administered via ultrasound-guided intracardiac injection into the left ventricle. Left ventricle administration was chosen for cells to enter the arterial circulation rather than the venous. This allows cells to pass through all organs before returning to the right ventricle and subsequently lungs, where they become sequestered in the pulmonary microvasculature (*Kean et al., 2013*; *Fischer et al., 2009*). To provide a detailed assessment of cell localisation, we combined photoacoustic tomography imaging at 1 mm steps with bioluminescence detection. While bioluminescence

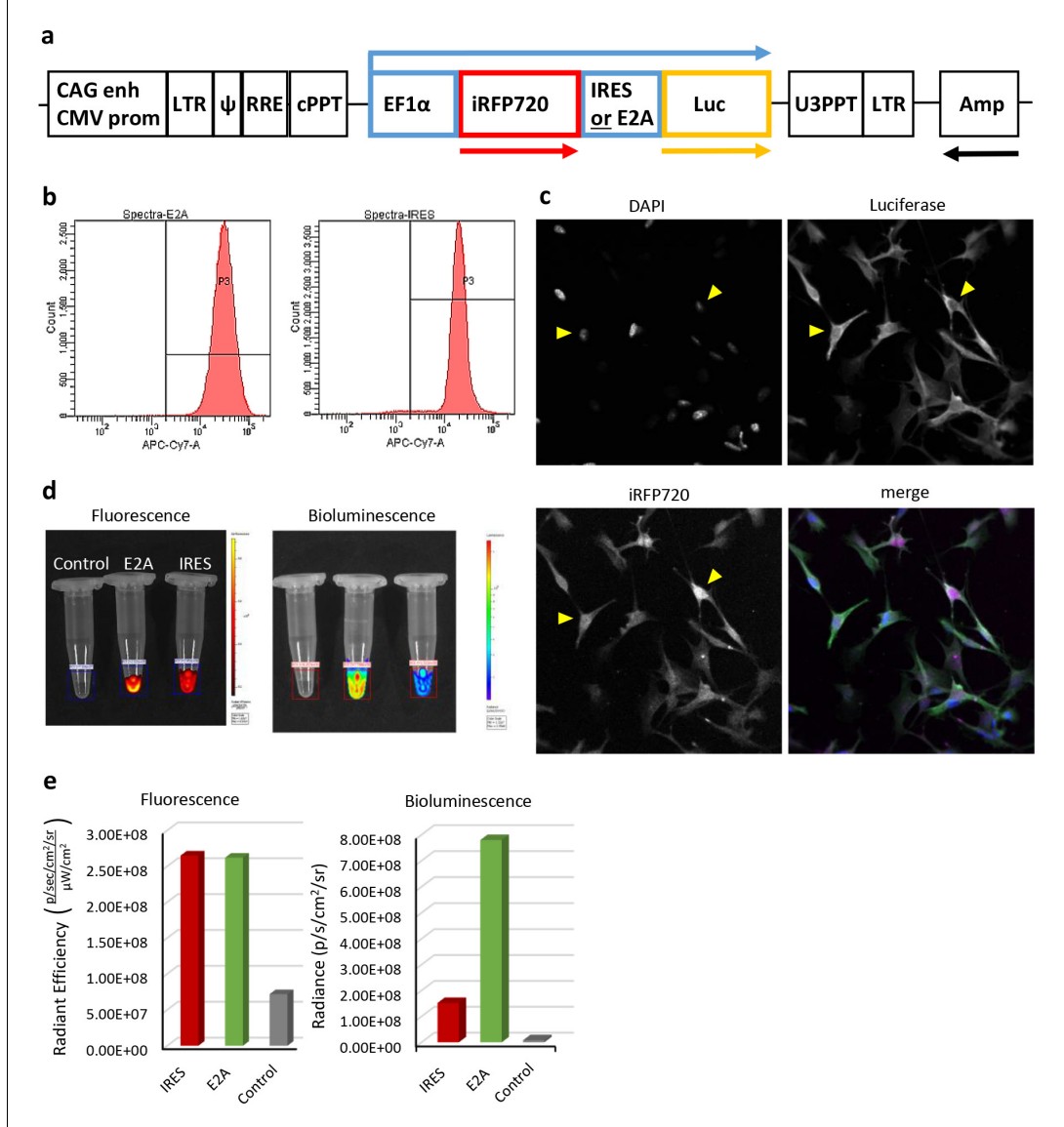

**Figure 2.** Characterisation of iRFP720-Luciferase bi-cistronic lentivirus reporter gene vectors and transfected cells in vitro. (a) Scheme of the vectors that mediate co-expression of iRFP720 and firefly luciferase either via IRES- or E2A-mediated mechanisms (not drawn to scale). Arrows above and below the scheme indicate the reporter gene transcript and open reading frames, respectively. (b) FACS analyses of iRFP720 fluorescent signal intensity in purified mMSC populations, which had been transfected with either the IRES or E2A vector, respectively, indicated similar levels of expression of the first ORF. (c) Confocal fluorescence microscopy of transfected cells for co-expression of iRFP720 and luciferase (immunofluorescence). In contrast to Luc, iRFP720 tends to accumulate in the nucleus (yellow triangles). (D, E) COMPARISON OF FLUORESCENT AND BIOLUMINESCENT SIGNAL INTENSITIES FROM IRES AND E2A VECTOR-TRANSFECTED CELL POPULATIONS USING IVIS SPECTRUM IMAGING. ALTHOUGH BOTH CELL SUSPENSIONS PROVIDED VERY SIMILAR LEVELS OF FLUORESCENCE FROM THE FIRST REPORTER ORF (iRFP720) IN THE REGION OF INTEREST (ROI), BIOLUMINESCENCE LEVELS WERE 5-FOLD HIGHER IN THE E2A VECTOR-TRANSFECTED CELL POPULATION, INDICATING A MORE EFFICIENT TRANSLATION OF THE SECOND ORF (LUCIFERASE) FROM THE E2A ELEMENT THAN FROM THE IRES. FLUORESCENCE WAS MEASURED FIRST, FOLLOWED BY ADDITION OF CYCLUC1 TO THE SAME CELL SUSPENSIONS AND BIOLUMINESCENCE IMAGING.

DOI: https://doi.org/10.7554/eLife.33140.005

The following figure supplement is available for figure 2:

**Figure supplement 1.** Longitudinal multi-modal imaging of iRFP720-E2A-Luc labelled cells after subcutaneous injection of mMSCs shows proportionality of bioluminescent, fluorescent and photoacoustic signals of the reporter genes.

DOI: https://doi.org/10.7554/eLife.33140.006

imaging provides excellent sensitivity, its anatomical resolution is poor, mainly due to the high

scattering of photons by tissue (*Patterson et al., 1989*; *Ntziachristos et al., 2005*). By contrast, photoacoustic imaging provides superior spatial resolution (150 μm in our system) (*Ntziachristos and Razansky, 2010*). To demonstrate the potential of this multimodal approach to precisely determine the localisation of distributed nanorod-labelled cells, we show the correlation between bioluminescence imaging and MSOT imaging (*Figure 3*). Cell localisation in the head, liver and kidney regions was clearly observed with both imaging modalities (*Figure 3a,b*), and the cell presence within organs was confirmed in the photoacoustic scans through the respective regions (*Figure 3c*). This distribution is in agreement with other, different types of assessments in previous reports (*Basse et al., 1988*). The presence of cells in the brain region is probably due to cell trapping in small capillaries, as reported also in other studies after intracardiac injection of cells (*Heyn et al., 2006*). As shown in our parallel work (*Scarfe et al., 2017*), the mMSCs remain in the lumen of the brain capillaries and do not cross the blood-brain barrier. It should be noted here that there was no difference in the bioluminescence signal biodistribution when cells with or without gold nanorods were injected, indicating that GNR uptake does not affect the overall behaviour of the cells (*Figure 3—figure supplement 1*).

To further test the robustness and specificity of the signal, additional controls were performed. First, following the same imaging protocols, we observed a similar biodistribution pattern for mice injected with either GNR-710 or GNR-860-labelled cells (*Figure 3—figure supplement 1*, and *Figure 3—figure supplement 2*). Second, when we applied the unmixing algorithm with the 'wrong' spectrum (i.e. the GNR-710 spectrum to an animal injected with GNR-860-labelled cells, and vice versa), no GNR-specific signals were detected in photoacoustic images pre- and post-injection (*Figure 4*). Third, when the photoacoustic spectra of regions of interest before and after injection of cells were extracted, an increase in the photoacoustic intensity was only observed in the range of wavelengths in which corresponding GNRs have the higher absorbance, while intensity in the other wavelengths remained similar (*Figure 4—figure supplement 1*). It has to be noted that in some cases there were regions with an endogenous absorbance similar to GNRs (e.g. food in the intestines has similar spectra as GNR-710), which can lead to misinterpretation (*Figure 3—figure supplement 2*). For this reason, a scan was performed before cells were injected, which allowed detection of any potential endogenous interference with the GNR signal. Using a multimodal imaging approach (in this case luminescence and photoacoustic) also assisted in discounting any false positive signals. As expected, the photoacoustic signal provided by iRFP720 expression was not strong enough to detect the cells immediately after injection by means of photoacoustic imaging (*Figure 4*).

## Clearance and fate of the cells and GNRs after initial biodistribution

One day after GNR-labelled cell infusion, only between 15–22% of the initial bioluminescence signal in whole mice persisted (*Figure 5b*), decreasing to less than 4% by day 5. Such substantial amounts of cell death following cell infusion were expected from previous reports (*Tögel et al., 2008*; *Bentzon et al., 2005*). A similar degree of cell death was observed with cells not labelled with GNRs, indicating that cell loss in vivo was not caused by GNR labelling (*Figure 5—source data 1*). Interestingly, after 24 hr, the bioluminescence signal and the photoacoustic signal for GNRs were no longer localised in the same regions/organs (*Figure 5a–c*), indicating that the vast majority of GNRs followed a different fate than the remaining luciferase-expressing cells. Specifically, although the luminescence intensity had decreased substantially by 24 hr, the distribution of the signal remained the same (*Figure 5b*). By contrast, the MSOT signal for GNR-860 increased dramatically in the liver over the same timescale, but was substantially reduced in regions such as the kidneys and head where it had been predominant immediately after cell injection (*Figure 5a,c*). Whilst luminescent signal is based on living cells, photoacoustic signal can be produced from GNRs independently, irrespective of whether they are within cells, or if the cells are viable. Thus, these data suggest that the liver plays an important role in the clearance of GNRs and potentially associated cell debris following the massive loss of cells during the first 24 hr after injection, as previously reported (*Eggenhofer et al., 2012*; *Khabbal et al., 2015*). In addition, these findings are in line with reports describing nanoparticle clearance through the hepato-biliary tract (*Longmire et al., 2008*; *Hirn et al., 2011*; *Lankveld et al., 2011*). We observed that the clearance could be continuously monitored in a longitudinal fashion since the photoacoustic signal in the liver dropped by 47% after 5 days and by 73% after 25 days (*Figure 5d*).

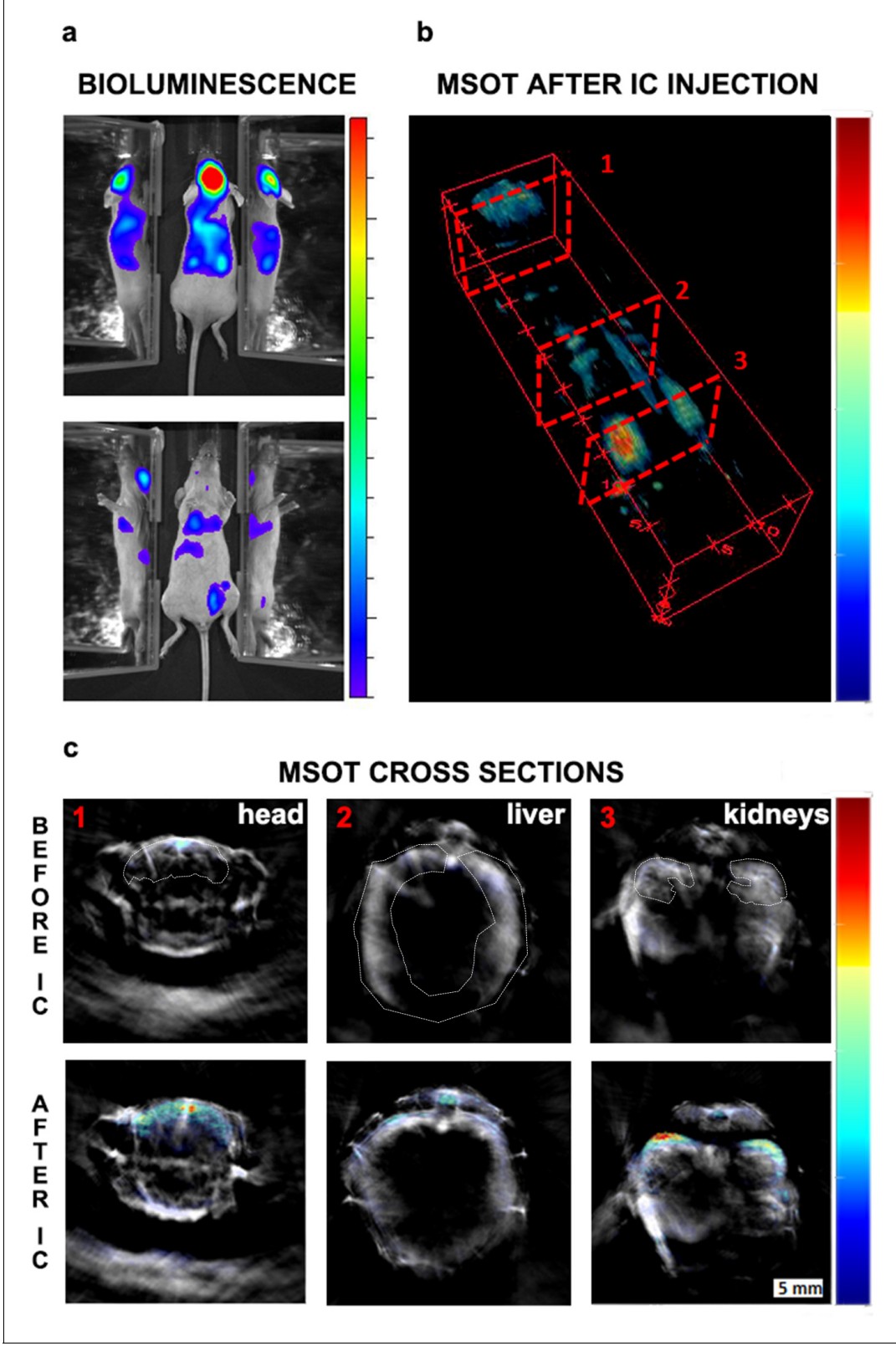

**Figure 3.** Initial biodistribution of GNR-860-labelled cells after intracardiac administration. (a) Bioluminescence imaging in dorsal (top panel) and ventral (bottom panel) position after cell administration. Mirrors were placed on the side of the mouse to provide a lateral view. (b) MSOT imaging of the same animal: 3D reconstruction for the whole animal showed the correlation between GNR-860 signal with bioluminescence imaging (360° animation available in *Figure 3—video 1*). (c) Cross sections corresponding to the planes shown in (b) before and after the injection of cells. Sub-organ
*Figure 3 continued on next page*

*Figure 3 continued*

localisation of cells can be observed here (see main text for details and source data at Zenodo (https://zenodo.org/record/1021607) (doi: 10.5281/zenodo.1021607) for the whole set of cross section images). The brain, liver and kidney are outlined. Colour scales are $3.0 \times 10^5$ to $1.8 \times 10^6$ radiance units ($\rho \cdot s^{-1} \cdot cm^{-2} \cdot sr^{-1}$) in (a) and 0.9 to 21 MSOT intensity (a.u.) in (b) and (c).

DOI: https://doi.org/10.7554/eLife.33140.007

The following video and figure supplements are available for figure 3:

**Figure supplement 1.** Biodistribution of cells not labelled with GNRs and cells labelled with GNR-710 and GNR-860, respectively, after intracardiac administration.

DOI: https://doi.org/10.7554/eLife.33140.008

**Figure supplement 2.** MSOT imaging of cell biodistribution of GNR-710 labelled cells – comparison before and after cell injection.

DOI: https://doi.org/10.7554/eLife.33140.009

**Figure 3—video 1.** 360° animation of the 3D reconstruction shown in *Figure 3* of the main text.

DOI: https://doi.org/10.7554/eLife.33140.010

## Tumour monitoring

Following the substantial loss in cell viability over the first 24 hr, signal intensity declined further and by day 5 cells were no longer detectable by BLI in the regions where they had first accumulated (i.e. brain, kidneys and spinal cord, *Figure 3*). However, we now observed BLI signals from cells that had settled in different locations (*Figure 6—figure supplement 1*). The signal intensity increased over the following weeks as cells grew into tumours (*Figure 6—figure supplement 1*). Since the mMSC cell line had been reported to form osteoid structures in vivo (*Kuzma-Kuzniarska et al., 2012*), and recent results from our group have revealed the capacity of the cells to form osteosarcomas (*Scarfe et al., 2017*), we anticipated the possibility of tumour development. In other studies, this process has been typically monitored by bioluminescence imaging (*Jenkins et al., 2005*), which allows the intensity of the signal to be correlated with the number of living cells. However, bioluminescence is limited by the poor spatial resolution at depths beyond 1 mm due to photon scattering. We therefore evaluated the potential of MSOT to determine the precise localisation of internal tumours as well as their size and shape. We used the constitutive expression of iRFP720 for the long-term monitoring of tumour growth by means of photoacoustic imaging. Presence of iRFP720 resulted in a change of the photoacoustic spectra in the regions where cells were forming tumours (*Figure 6*, and *Figure 6—figure supplement 2*). The fact that the iRFP720 absorption spectrum is very different from the main endogenous absorbers (a sharp band peaking at 690–700 nm, *Figure 6—figure supplement 2*) facilitated its detection even with minimal changes of photoacoustic intensity compared to the background.

Mice developed tumours in different locations whether the cells were labelled with GNRs or not (*Figure 6* without GNRs and Figure 8 with GNRs). This process could be monitored longitudinally by bioluminescence and MSOT as shown in *Figure 6*, where a number of tumours formed in a mouse 30 days after administration of cells labelled with the iRFP720-Luciferase vector. After scanning the animal in 1 mm steps, multispectral processing of the corresponding cross sections (*Figure 6b*) revealed the presence of tumours on the right shoulder (1), spinal cord (2), in the region dorsal of the kidney (2a and 2b), back (3), hip (4 and 5), and right leg (6). Tumours 2a and 2b are very illustrative of the outcomes that can be achieved with this approach. From the in vivo bioluminescence images (*Figure 6a*), it is difficult to determine the localisation of the tumours as either subcutaneous or in deep tissues, and with which organs they are associated. Analysis of photoacoustic images revealed that those tumours were growing in an unusual position close to and dorsal of the kidneys (*Figure 6b*). *Ex vivo* luminescence imaging confirmed the observations made with *in vivo* photoacoustic tomography (e.g. tumour masses 2a and 2b were found attached to the anterior-dorsal part of the kidneys) (*Figure 6c*). Expression of iRFP720 was also confirmed by *ex vivo* fluorescence imaging of these tumours (*Figure 6—figure supplement 3*).

In addition to improving the anatomical localisation of internal tumours, photoacoustic tomography using iRFP720 expression also allowed precise longitudinal monitoring of tumour development from very early stages. The example in *Figure 7* shows the growth of tumour 4 (*Figure 6*) from day 13 to day 30 after cell injection. The excellent spatial resolution enabled the localisation of this tumour in the vicinity of the pelvic ilium from day 13. The tumour mean diameter at this stage was 1.2 mm. By day 19, 23 and 30 it had expanded to 2.0, 2.2 and 3.3 mm, respectively. Furthermore, changes in the shape

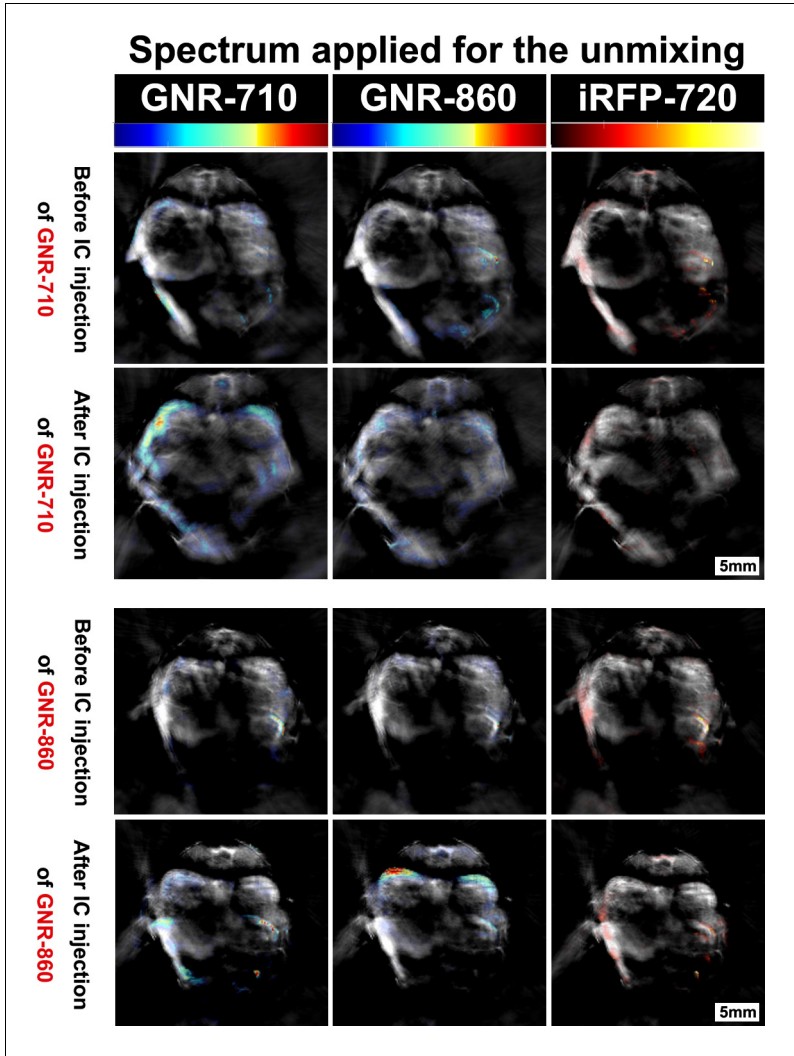

**Figure 4.** Specificity of the multispectral unmixing. Signal before and after treatment with GNR-710-labelled cells (upper panel) and GNR-860-labelled cells (bottom panel) in transverse sections at the level of the kidney region of the corresponding mouse. Signals were observed in the kidney only when the spectra of the respective component (GNR-710 in upper panel and GNR-860 in bottom panel) was applied. Note also that iRFP720 unmixing did not result in a signal at this stage. Colour scales are 0.1 to 14, 0.9 to 21, and 0.0 to 15 MSOT intensity (a.u.) for GNR-710, GNR-860, and iRFP720, respectively.

DOI: https://doi.org/10.7554/eLife.33140.011

The following figure supplement is available for figure 4:

**Figure supplement 1.** Photoacoustic spectra extracted from *in vivo* imaging.

DOI: https://doi.org/10.7554/eLife.33140.012

of the tumour were also monitored from day 23 onwards when it branched off into two ramifications, which grew attached to the main part of the tumour (*Figure 7* and *Figure 7—video 1*).

In a second example, the presence of several tumours was determined 40 days after injection of GNR-860 and reporter gene-labelled cells (*Figure 8*). Although GNRs were not used as contrast agents for tumour imaging, since the vast majority of them were cleared out following cell death, tumour growth was monitored with high spatial resolution via iRFP720 expression. The mouse shown in *Figure 8* developed tumours in both shoulders, in the right dorsal kidney/adrenal gland region, towards the liver, and in several positions of the hip region. A late developing (days 33–40), but fast-growing tumour in the left shoulder was used to demonstrate the capabilities of iRFP720 MSOT imaging to obtain detailed spatial information in 3D reconstructions as shown in *Figure 8c*.

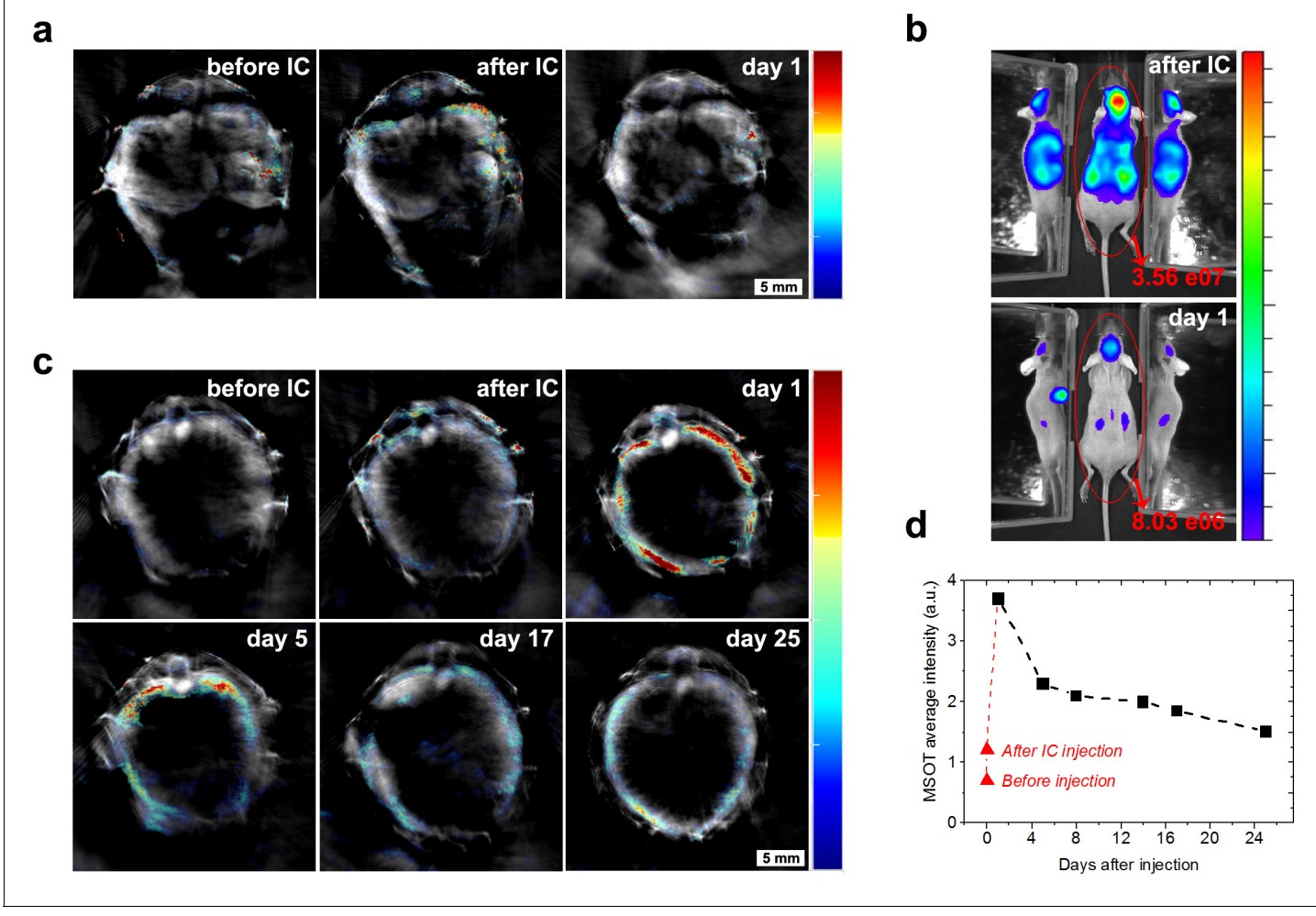

**Figure 5.** Cell death and clearance of GNR-860 from day one after injection. (**a**) MSOT signal in kidneys disappeared almost completely at day 1. (**b**) Total bioluminescence decreased by 78% at day one indicating a large amount of cell death. (**c**) In contrast to the kidneys, MSOT signal in the liver increased substantially at day 1, although bioluminescence had declined in this tissue at this stage as shown in (**b**), which suggests that GNRs accumulated in the liver. (**d**) After day 1, MSOT signal in liver decreased progressively. Colour scales in (**a**) and (**c**) are 0.5 to 8 MSOT intensity (a.u.). Colour scale in (**b**) is $7.0 \times 10^4$ to $5.8 \times 10^5$ radiance units ($\rho \cdot s^{-1} \cdot cm^{-2} \cdot sr^{-1}$).

DOI: https://doi.org/10.7554/eLife.33140.013

The following source data is available for figure 5:

**Source data 1.** Loss in BLI signal intensity within 24 hr after injection.
DOI: https://doi.org/10.7554/eLife.33140.014

Overall, our analysis reveals that MSOT provides an excellent method to monitor tumour growth longitudinally from very early stages onwards (from sub mm size). Only some tumours growing in very peripheral positions of the animal could not be monitored due to technical or procedural limitations of the MSOT imaging system. For example, the nose could not be imaged as it was placed in the nose cone for air and anaesthesia supply during the imaging process. Also, since the animal was placed on its back, it was very difficult to avoid an air layer being trapped between the legs and the main body, which impaired sound propagation. Abundant use of ultrasound gel in this region minimised this effect and allowed data to be obtained as shown in *Figure 6* (tumour 6), although the quality of the image was not as good as in other regions of the body. Any tumours in the lungs could not have been monitored with photoacoustic imaging due to a different sound propagation in air. However, the presence of tumours in the lungs could be ruled out from the bioluminescence images.

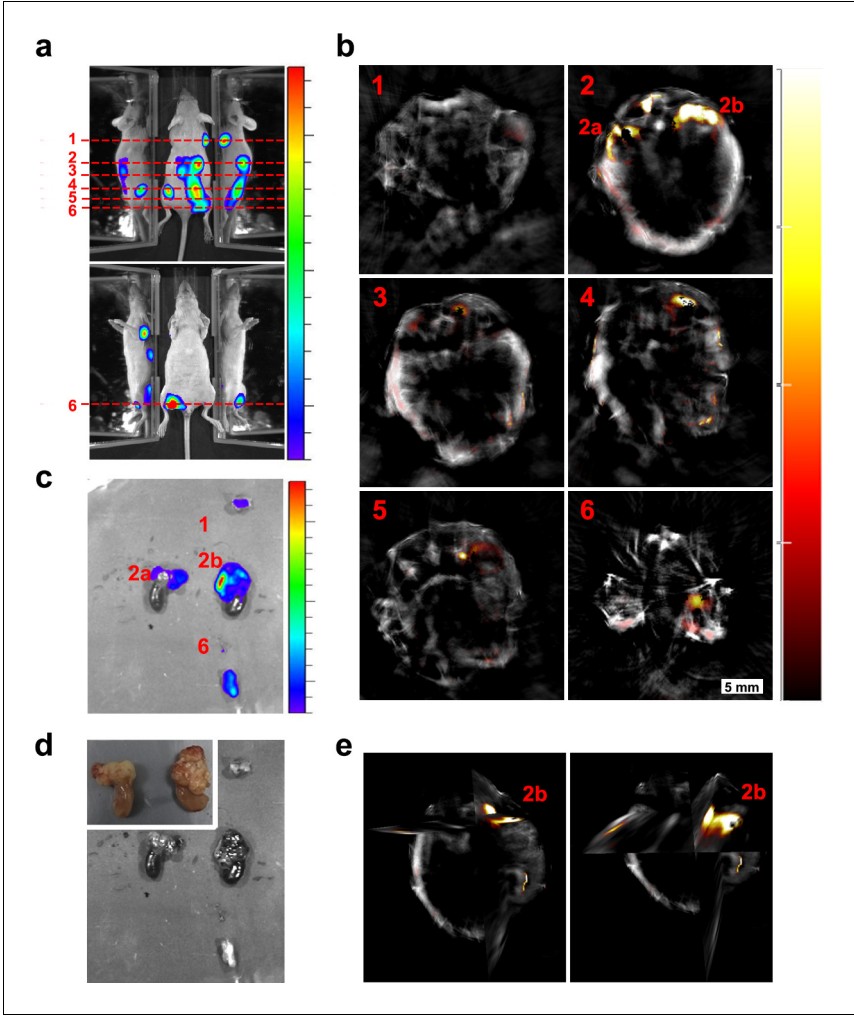

**Figure 6.** Localisation of tumours 30 days after injection of cells. (**a**) Bioluminescence images showing different tumours spread along the mouse body. Dashed lines indicate the approximate region shown in the MSOT scans. (**b**) Corresponding MSOT transverse sections showing the precise localisation of different tumours after applying the multispectral processing for iRFP720. (**c**) *Ex vivo* bioluminescence imaging confirmed the localisation of relevant tumours imaged with MSOT. (**d**) *Ex vivo* picture of the tumours shown in (**c**) with a detail of kidneys in the inset. (**e**) Orthoslice images of tumour 2b in (**b**). Frontal view (left) and tilted view (right) provide a better spatial visualisation of the tumour´s precise localisation, extending anterior from the dorsal part of the right kidney. Colour scales are $1.4 \times 10^8$ to $8.5 \times 10^8$ radiance units ($\rho \cdot s^{-1} \cdot cm^{-2} \cdot sr^{-1}$) in (**a**) and $2.6 \times 10^8$ to $5.3 \times 10^9$ radiance units in (**c**). Colour scale in (**b**) is 0.4 to 15 MSOT intensity units (a.u.).
DOI: https://doi.org/10.7554/eLife.33140.015

The following figure supplements are available for figure 6:

**Figure supplement 1.** Bioluminescence monitoring of tumour growth of the mouse shown in *Figures 6* and *7* of the main text.
DOI: https://doi.org/10.7554/eLife.33140.016

**Figure supplement 2.** *In vivo* photoacoustic spectra of tumour 2b in *Figure 6* of the main text.
DOI: https://doi.org/10.7554/eLife.33140.017

**Figure supplement 3.** *Ex vivo* epi-fluorescence image (Ex = 675 nm, Em = 720 nm) of tumours shown in *Figure 6*, after fixation with 4% PFA for 24 hr.
DOI: https://doi.org/10.7554/eLife.33140.018

## Conclusions

We demonstrate here a multimodal imaging approach, which utilises a combination of GNRs and reporter genes to track cells immediately after systemic injection and longitudinally over time during

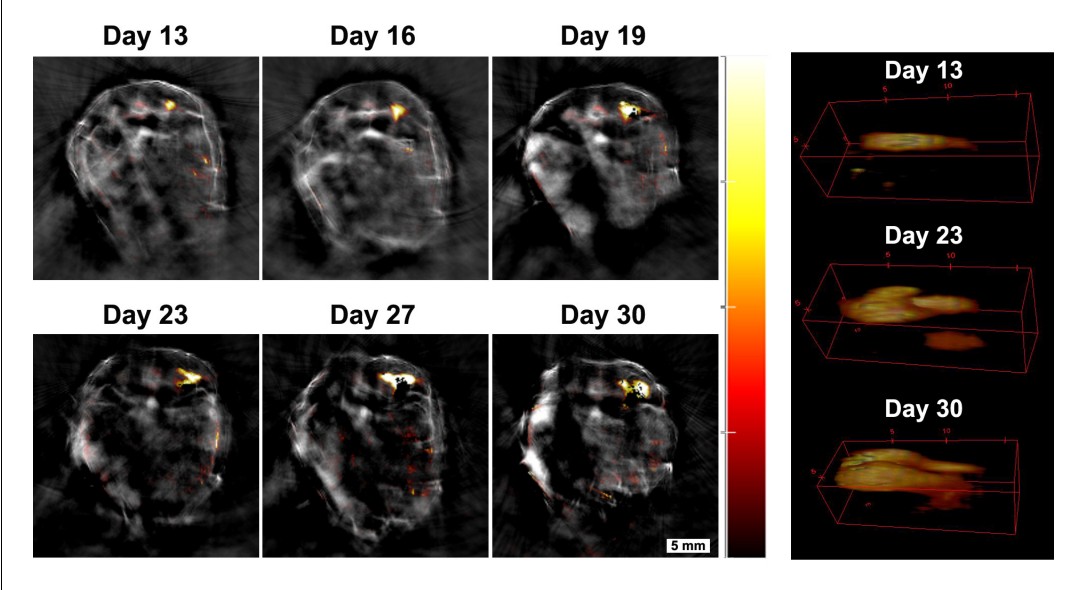

**Figure 7.** Monitoring tumour growth over time. Tumour four in *Figure 6* was chosen to demonstrate the potential of the MSOT approach in monitoring tumour growth longitudinally over time. A small tumour was localised in the pelvic ilium region at day 13 and its growth was monitored from day 13 to day 30. 3D reconstructions of the tumour are shown on the right panel (360° animations are available as *Figure 7—video 1*). Changes in size can be determined with this technique. Colour scale is 0.4 to 15 MSOT intensity units (a.u.).

DOI: https://doi.org/10.7554/eLife.33140.019

The following video is available for figure 7:

**Figure 7—video 1.** 360° animation of the 3D reconstructions shown in *Figure 7* of the main text.

DOI: https://doi.org/10.7554/eLife.33140.020

the development of tumours. We show that multispectral optoacoustic tomography achieves high spatial resolution of the initial cell distribution through analysis of the GNR signal. Whilst expression of iRFP720 is initially undetectable, due to the scattered distribution of cells, it provides long-term tumour monitoring capability in MSOT as clustered iRFP720-positive cells lead to a stronger photoacoustic signal. Additionally, bioluminescence imaging of luciferase allowed for the most sensitive detection of live cells for comparison and validation of the MSOT results.

The silica-coated GNRs provided the excellent photoacoustic sensitivity required to monitor the broad distribution of labelled cells shortly after systemic intra-cardiac injection. Because the GNRs maintained their optical signature when taken up by the cells, spectral unmixing could be applied in MSOT, which allowed GNR signals to be distinguished from other reporters (e.g. iRFP720) and endogenous absorbers, resulting in highly specific cell detection. Over the medium term, that is several days post-injection, photoacoustic GNR signals and luciferase bioluminescence signals became separated, indicating nanoparticle clearance and surviving cells, respectively. The GNR signal became localised to the liver, which is in line with the known clearance pathway of nanoparticles and associated cell debris through the hepato-biliary system.

Using the genetic reporters, longitudinal monitoring of surviving cells and tumour formation was achieved via luciferase bioluminescence and iRFP720 photoacoustic tomography. Compared to other optical-based imaging technologies, the superior deep-tissue spatial resolution of MSOT provided precise information about anatomical localisation and morphology of tumours from very early stages (<1 mm diameter).

Beyond the specific study of the biodistribution and tumour development of an exemplary mesenchymal cell line performed here, we emphasise the technological potential of these labelling tools for non-invasive *in vivo* imaging and cell tracking. In particular, research fields such as cell-based regenerative medicine therapies and cancer biology might find new applications using these labelling reagents and imaging approaches for cell monitoring and safety studies.

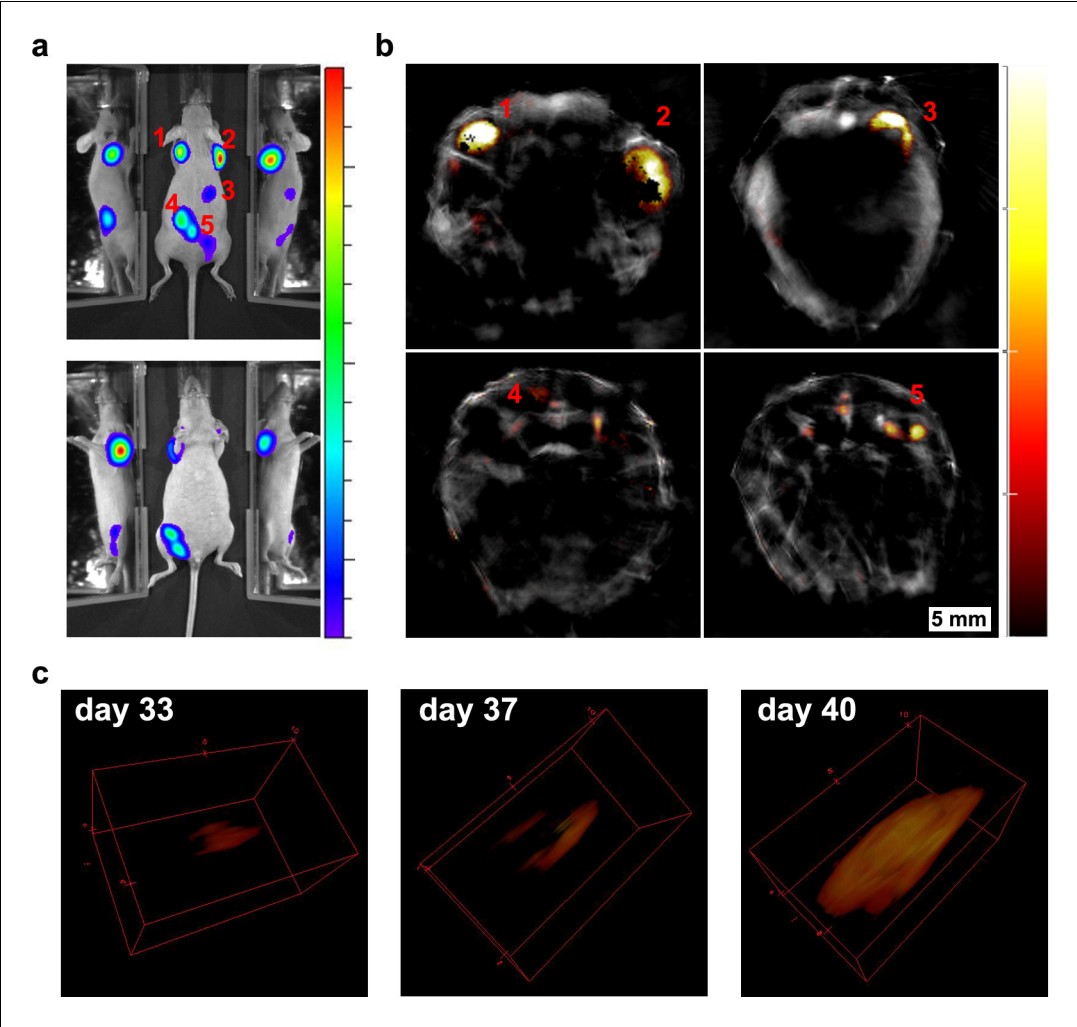

**Figure 8.** Longitudinal tumour monitoring in a mouse that was injected with GNR-860/reporter gene labelled cells. Localisation of tumours 40 days after systemic injection of cells imaged by bioluminescence (**a**) and MSOT (**b**). (**c**) Growth of the tumour on the left shoulder was monitored from day 33 to day 40 and is shown in 3D reconstruction. Colour scale in (**a**) is $1.4 \times 10^8$ to $1.3 \times 10^9$ radiance units ($\rho \cdot s^{-1} \cdot cm^{-2} \cdot sr^{-1}$) and in (**b**) is 1.2 to 14 MSOT intensity units (a.u.). The black pixels visible within the tumour region in (**b**) are an experimental artefact (see *Figure 8—figure supplement 1* for additional discussion).

DOI: https://doi.org/10.7554/eLife.33140.021

The following figure supplement is available for figure 8:

**Figure supplement 1.** Comparison of MSOT image reconstruction using software version 3.6 and 3.8.

DOI: https://doi.org/10.7554/eLife.33140.022

# Materials and methods

**Key resources table**

| Reagent type (species) or resource | Designation | Source or reference | Identifiers | Additional information |
|---|---|---|---|---|
| SCID hairless outbred (SHO) mice | mice | Charles River Laboratories | ID_Crl:SHO-Prkdc$^{scid}$Hr$^{hr}$ | |
| Mouse mesenchymal stem cell line | mMSC | ATCC | ID_ATCC:CRL-12424 | |

*Continued on next page*

*Continued*

| Reagent type (species) or resource | Designation | Source or reference | Identifiers | Additional information |
|---|---|---|---|---|
| Lentivirus vector | pHIV-iRFP720-E2A-Luc | this paper | ID_GenBank:MF693179; ID_Addgene:104587 | lentivirus vector for bi-cistronic expression of iRFP720 and firefly luciferase |
| Gold nanorods | GNR | this paper | | gold nanorods were produced at the Department of Chemistry, University of Liverpool |

## Reagents

The following chemicals were purchased from Sigma-Aldrich (Gillingham, UK). $HAuCl_4 \cdot 3H_2O$ (>99.0%), $NaBH_4$ (>99.99%), $AgNO_3$(99.0%), hexadecyltrimethylammonium bromide (CTAB,>99%), L-ascorbic acid (reagent grade), tetraethyl orthosilicate (TEOS, 99.999%), O-[2-(3-Mercaptopropiony-lamino)ethyl]-O′-methylpolyethylene glycol (mPEG-SH, MW: 5000 Da), 5-bromosalicylic acid. Mono-carboxy (1-mercaptoundec-11-yl) hexaethylene glycol (PEG-COOH, MW 526.73 Da) was obtained from Prochimia (Sopot, Poland). Dulbecco's modified eagle's medium (DMEM), phosphate-buffered saline (PBS), penicillin-streptomycin, biliverdin and polybrene were also obtained from Sigma Aldrich. Foetal bovine serum (FBS) was purchased from Life-Technologies. D-Luciferin was purchased from Promega (Southampton, UK) and CycLuc1 from Glixx Laboratories (Southborough, USA).

## Gold nanorod synthesis

GNR synthesis were based on previously published protocols (*Nikoobakht and El-Sayed, 2003*; *Ye et al., 2012*). First, seeds were prepared by adding 0.6 mL of ice-cold $NaBH_4$ (0.01 M) to a mixture of 5 mL CTAB (0.2 M) and 5 mL of $HAuCl_4$ (0.5 mM) under vigorous stirring. The growth solution for GNR-710 was prepared by mixing 50 mL of CTAB (0.2M), 1.8 mL of $AgNO_3$ (4 mM), 50 mL of $HAuCl_4$ (1 mM) and 0.8 mL of ascorbic acid (0.1 M). Finally, 0.2 mL of freshly synthesised seeds was added to the growth solution. The reaction was kept in a water bath at 28°C for 3 hr. The growth solution for GNR-860 was prepared by adding 440 mg of 5-bromosalicylic acid to 100 mL of CTAB (0.1 M) and heat up the mixture to 60°C. After cooling down the solution to 30°C, 4.8 mL of $AgNO_3$ (4 mM) was added and left undisturbed for 15 min. Then, 50 mL of $HAuCl_4$ (1 mM) was added and left under low stirring for 15 min at 30°C. Afterwards, 256 μL of ascorbic acid was added under vigorous agitation for 30 s. Finally, 160 μL of seeds were added and the solution was kept at 28°C for 12 hr. Silica coating was performed as described in our previous work (*Comenge et al., 2016*), which was based on slight modifications of a protocol by Fernández-López et al (*Fernández-López et al., 2009*).

GNRs were visualized using a Tecnai G3 Spirit transmission electron microscope (TEM) at 120 keV. Formvar/carbon-coated 200 mesh copper grid (TAAB) were dipped in a solution of the GNRs of interest and left to dry in air. More than 100 GNRs were considered for image analysis.

## Lentivirus vectors

Lentivirus vectors were constructed on the pHIV-Luciferase backbone (a gift from Bryan Welm, Addgene plasmid # 21375). The iRFP720 ORF was obtained from plasmid piRFP720-N1 (a gift from Vladislav Verkhusha, Addgene plasmid # 45461) (*Shcherbakova and Verkhusha, 2013*) and cloned as an EcoRI/XbaI fragment upstream of the IRES element of pHIV-Luciferase, resulting in the pHIV-iRFP720-IRES-Luc vector. To achieve a more efficient translation of the second ORF, the IRES element was replaced with an E2A peptide motif (*Szymczak and Vignali, 2005*; *Szymczak et al., 2004*) using a two-step PCR protocol to create the pHIV-iRFP720-E2A-Luc plasmid. The complete sequence of the E2A plasmid version has been submitted to the NCBI GenBank database (accession number: MF693179). Lentivirus particles were produced in HEK293T cells, which were purchased for this work from ATCC (ATCC CRL-3216) and free of mycoplasma when tested with the e-Myco Mycoplasma PCR Detection Kit (iNtRON Biotechnology, cat. no. 25235). Cells were transfected via calcium phosphate co-transfection of the reporter gene plasmid, packaging plasmid psPAX2 and envelope plasmid pMD2.G (both a gift of Didier Trono, Addgene plasmids #12260 and #12259,

respectively) as described (*Kutner et al., 2009*). Virus-containing cell culture supernatant was collected three days post-transfection, centrifuged at 500 g and filtered through 0.45 μm pores.

## Mouse mesenchymal stem/stromal cell (mMSC) line culture, labelling and immunofluorescence

We chose the mouse MSC line D1 ORL UVA [D1](ATCC CRL-12424) for labelling and tracking, since these cells have the potential to differentiate into mesenchymal derivatives, but can occasionally also form tumours as previously shown (*Comenge et al., 2016*; *Kuzma-Kuzniarska et al., 2012*). The mMSC cell line was purchased for this work from ATCC and free of mycoplasma when tested with the e-Myco Mycoplasma PCR Detection Kit (iNtRON Biotechnology, cat. no. 25235). Furthermore, the mMSC cells were re-authenticated by Science Exchange – IDEXX BioResearch using short tandem repeat (STR) profiling. They were found to be of mouse origin and no other mammalian interspecies contamination was detected. They were grown in DMEM supplemented with 10% FBS and 2 mM L-Glutamine. Cells were transfected with purified reporter gene virus particles for 24 hr. iRFP720-expressing cells were sorted using a BD FACSAria III with red laser and APC-Cy$^{TM}$7 filter. Cells expressing high levels of iRFP720 were selected and maintained for all further experiments. Before IVIS imaging of cells, biliverdin was added to the culture medium for 24 hr to increase its concentration beyond levels present in FBS and to improve its uptake by cells for incorporation into iRFP720 as a chromophore. For analysis of reporter gene expression by immunofluorescence, mMSCs were grown on glass cover slips, fixed with 4% paraformaldehyde/PBS, washed with PBS, blocked and stained with an anti-firefly luciferase antibody (Abcam, Cambridge, UK, ab21176, diluted in 1:500 in PBS/10% donkey serum/0.25% Triton-X100) and a donkey-anti-rabbit-Alexa-Fluor488 secondary antibody (Molecular Probes, A21206). Fluorescence of iRFP720 was assessed directly using a Zeiss LSM510 Multiphoton microscope.

GNR-labelling was performed as previously published (*Comenge et al., 2016*). Briefly, cells were treated with cell medium containing GNRs at the concentration/optical density indicated in the Results (always 79% cell medium, 20% GNRs in water, 1% penicillin-streptomycin). Unless otherwise stated, cells were labelled to a GNR O.D. = 3.0. Then, cells were dissociated with trypsin, resuspended in fresh medium, washed twice with PBS, and counted using an automated cell counter (TC10, BioRad, Watford, UK).

## Cell viability

Cell viability was assessed with Cell Titer Glo ATP Assay (Promega). Cells were labelled as described above by seeding $10^4$ mMSCs cells in 96-well plates. Cells were incubated with GNRs-710 and GNR-860 at a final O.D. in media = 1, 2, 3.2, and 4 (80% media and 20% GNRs solution in each case) for 24 hr. After labelling, cells were washed three times with PBS. 50 μL of medium were added to each well and then 25 μL of the ATP reagent was added. The plate was mixed in an orbital shaker and, after 10 min, the contents of the plate were transferred to white, opaque, 96-well plates and the luminescence measured with a plate reader (Fluostar Omega, BMG Labtech, Aylesbury, UK). Each condition was assessed in triplicate and results are given as %±SD relative to cells that were incubated without GNRs as described above.

## Preparation of cells for TEM

Cells were fixed with a solution containing 1% paraformaldehyde and 3% glutaraldehyde in 0.1 M cacodylate buffer (pH 7.4). Then, cells were incubated with a reduced osmium staining solution, containing 2% $OsO_4$ and 1.5% $K_4[Fe(CN)_6]$, for 1 hr. This was followed by a second 1 hr osmium staining (2% $OsO_4$) step and overnight staining with 1% uranyl acetate. Cells were washed with water for 3 min, three times after every staining step. Samples were then dehydrated in graded ethanol (30%, 50%, 70%, 90% and 2 × 100%) for 5 min each. Finally, samples were infiltrated with medium TAAB resin 812 and embedded within the same resin. The resin was cured for 48 hr at 60°C. Finally, ultrathin sections of 350 μm x 350 μm x 74 nm were cut and placed in 200-mesh Formvar/Carbon filmed grids. They were post-stained with uranyl acetate (4% UA in a 50:50 ethanol/water solution) and Reynolds lead citrate before TEM imaging.

**Table 1.** Summary of mice used and analysed for the experiment.

| Group | Transfected | GNR-710 | GNR-860 | Number of animals |
|---|---|---|---|---|
| I (control) | ✓ | | | 2 |
| II (GNR-710) | ✓ | ✓ | | 3 |
| III (GNR-860) | ✓ | | ✓ | 4 |

DOI: https://doi.org/10.7554/eLife.33140.023

## Animals

8–10 week-old female SCID hairless outbred (SHO) mice (Charles River, Margate, UK) were housed in individually ventilated cages at a 12 hr light/dark cycle, with *ad libitum* access to food and water. Experimental animal protocols were performed in accordance with the guidelines under the Animals (Scientific Procedures) Act 1986 (licence PPL70/8741) and approved by the University of Liverpool Animal Welfare and Ethical Review Body. The tumour burden was monitored and kept within recommended limits in accordance with guidelines for the welfare and use of animals in cancer research (*Workman et al., 2010*). Experiments are reported in line with the ARRIVE guidelines. These experiments aimed at evaluating the potential of MSOT to track cells over the short and long-term. The number of animals was chosen so that a range of tumour positions and sizes could be observed (*Table 1*). Image quantification is presented to demonstrate the information that can be extracted from this approach. In each case, the numbers correspond to the particular animal presented. The data for these exemplar animals as well as for the other animals are available in the data repository Zenodo (*Comenge et al., 2017*).

## Intracardiac injection and imaging

Transfected cells were labelled with GNR-710 and GNR-860 at O.D. = 2.4 as detailed above. $1.5 \times 10^6$ cells in 100 µL PBS were prepared for intracardiac injection as described below. All cell injection and imaging procedures of mice were carried out under general isoflurane/oxygen anaesthesia.

The imaging routine at day 0 was composed of the following sequence of steps: 1) a baseline MSOT scan; 2) the ultrasound-guided (Prospect imaging system, S-Sharp, New Taipei City, Taiwan) intracardiac injection of cells into the left ventricle; 3) a second MSOT scan after 10 min for acclimatisation; 4) BLI, for which the mouse received an intraperitoneal injection of D-luciferin (approximately 150 µg/g body weight) 15 min before whole body imaging using an IVIS Spectrum system (Perkin Elmer, Seer Green, UK). All data were analysed with Living Image (Perkin Elmer) and data are displayed in radiance units. On subsequent days, the MSOT scan and BLI were repeated in the same manner.

For intracardiac injection of cells, mice were positioned supine on a heated platform. Fur around the chest area was removed using depilatory cream and limbs were taped down to keep the mouse position fixed. Ultrasound gel was applied liberally to the chest area and the ultrasound transducer (Prospect imaging system, S-Sharp) was positioned above the chest so the long axis view of the left ventricle was visible. 100 µl of cell suspension was drawn up into an insulin syringe (29 G) and, using the ultrasound image as guidance, was inserted into the left ventricle of the heart. Cell suspension was then administered slowly over a period of approximately 30 s.

For optoacoustic imaging, an MSOT inVision 256-TF small animal imaging system (iThera Medical GmbH, Munich, Germany) was used (*Morscher et al., 2014*). After acclimatisation for 10 min inside the water bath, a whole-body scan was performed on the mouse with 1 mm steps and the following wavelengths for acquisition: 660, 670, 680, 690, 700, 705, 710, 715, 720, 725, 735, 750, 765, 780, 795, 810, 820, 830, 840, 850, 860, 870, 880, 890, 900, 910, 920, 930, 940, 1025, 1050, 1075, and 1100 nm. Heavy water was used in the water bath due to its low absorbance at wavelengths > 910 nm (contrary to regular water). Linear-mode-based reconstruction and guided ICA multispectral processing were applied using viewMSOT software v3.6 (iThera Medical GmbH).

## Acknowledgements

We thank Joseph Zeguer and Danielle Vaughan for contributions to the lentivirus vector work, the Biomedical Services Unit, the Biomedical Electron Microscopy Unit and the Centre for Preclinical

Imaging at The University of Liverpool for technical support. We also acknowledge Marco Marcello and the Liverpool Centre for Cell Imaging (CCI) for provision of imaging equipment and technical assistance. Furthermore, we would like to thank B. Kevin Park for helpful discussion on the direction of the project.

## Additional information

### Competing interests

Joan Comenge: Joan Comenge is currently affiliated with Nanotargeting S.L. The research was conducted when the author was at the University of Liverpool, Institute of Integrative Biology. The author has no financial competing interests to declare. The other authors declare that no competing interests exist.

### Funding

| Funder | Grant reference number | Author |
| --- | --- | --- |
| Medical Research Council | | Joan Comenge<br>Jack Sharkey<br>Bettina Wilm<br>Mathias Brust<br>Patricia Murray<br>Raphael Levy<br>Antonius Plagge |
| Seventh Framework Programme | | Patricia Murray |
| UK Regenerative Medicine Platform | MR/K026739/1 | Joan Comenge<br>Jack Sharkey<br>Oihane Fragueiro<br>Bettina Wilm<br>Mathias Brust<br>Patricia Murray<br>Raphael Levy<br>Antonius Plagge |
| H2020 Marie Skłodowska-Curie Actions | NANOSTEMCELLTRACKING | Joan Comenge |
| Biotechnology and Biological Sciences Research Council | | Joan Comenge<br>Jack Sharkey<br>Bettina Wilm<br>Mathias Brust<br>Patricia Murray<br>Raphael Levy<br>Antonius Plagge |
| Engineering and Physical Sciences Research Council | | Joan Comenge<br>Jack Sharkey<br>Bettina Wilm<br>Mathias Brust<br>Patricia Murray<br>Raphael Levy<br>Antonius Plagge |

The funders had no role in study design, data collection and interpretation, or the decision to submit the work for publication.

### Author contributions

Joan Comenge, Resources, Data curation, Formal analysis, Validation, Investigation, Visualization, Methodology, Writing—original draft, Writing—review and editing; Jack Sharkey, Data curation, Formal analysis, Validation, Investigation, Visualization, Methodology, Writing—review and editing; Oihane Fragueiro, Resources, Data curation, Investigation, Methodology; Bettina Wilm, Investigation, Methodology, Project administration, Writing—review and editing; Mathias Brust, Resources, Supervision, Investigation, Methodology; Patricia Murray, Conceptualization, Supervision, Funding

acquisition, Investigation, Methodology, Writing—review and editing; Raphael Levy, Conceptualization, Resources, Supervision, Funding acquisition, Validation, Investigation, Methodology, Writing—original draft, Writing—review and editing; Antonius Plagge, Conceptualization, Formal analysis, Supervision, Validation, Investigation, Visualization, Methodology, Writing—original draft, Writing—review and editing

## Author ORCIDs
Bettina Wilm (iD) http://orcid.org/0000-0002-9245-993X
Mathias Brust (iD) http://orcid.org/0000-0001-6301-7123
Patricia Murray (iD) http://orcid.org/0000-0003-1316-148X
Raphael Levy (iD) http://orcid.org/0000-0001-5728-0531
Antonius Plagge (iD) http://orcid.org/0000-0001-6592-1343

## Ethics

Animal experimentation: Experimental animal protocols were performed in accordance with the guidelines under the Animals (Scientific Procedures) Act 1986 (licence PPL70/8741) and approved by the University of Liverpool Animal Welfare and Ethical Review Body. The tumour burden was monitored and kept within recommended limits in accordance with guidelines for the welfare and use of animals in cancer research. Experiments are reported in line with the ARRIVE guidelines. All cell injection and imaging procedures of mice were carried out under general isoflurane/oxygen anaesthesia.

## Decision letter and Author response

Decision letter https://doi.org/10.7554/eLife.33140.028
Author response https://doi.org/10.7554/eLife.33140.029

# Additional files

## Supplementary files

• Transparent reporting form
DOI: https://doi.org/10.7554/eLife.33140.024

## Major datasets

The following previously published dataset was used:

| Author(s) | Year | Dataset title | Dataset URL | Database, license, and accessibility information |
| --- | --- | --- | --- | --- |
| Joan Comenge, Jack Sharkey, Oihane Fragueiro, Bettina Wilm, Mathias Brust, Patricia Murray, Raphael Levy, Antonius Plagge | 2017 | Data: multimodal cell tracking from systemic administration to tumour growth by combining gold nanorods and reporter genes | https://doi.org/10.5281/zenodo.1021607 | Zenodo, Creative Commons Attribution 4.0, public |

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
