## [Decision Letter]

Thank you for submitting your article "Multimodal cell tracking from systemic administration to tumour growth by combining gold nanorods and reporter genes" for consideration by *eLife*. Your article has been reviewed by two peer reviewers, and the evaluation has been overseen by a Reviewing Editor and Fiona Watt as the Senior Editor. The following individual involved in review of your submission has agreed to reveal his identity: Jesse Jokerst (Reviewer #2).

The reviewers have discussed the reviews with one another and the Reviewing Editor has drafted this decision to help you prepare a revised submission.

Summary:

In their manuscript submitted to *eLife*, Comenge et al. describe a new way of combining genetic reporters and optoacoustic tomography to track grafted tumour-forming cells in mice. Their experimental system uses a viral vector which expresses both luciferase and a near infrared fluorescence protein, and they co-label these cells with gold nanorods (GNR). The GNRs allow the detection of small numbers of cells soon after grafting, whereas the fluorescent reporter signal becomes stronger with increasing cells numbers and enables Comenge et al. to track grafted cells by MSOT with a resolution superior to bioluminescence. The authors demonstrate the feasibility of their technology in mice injected with a mesenchymal stem cell line, which develops detectable tumours 2-3 weeks post-injection.

The study is of high technical quality, and the technical tools and methods reported here could be very useful for several research areas that use in vivo imaging, such as experimental cell therapy. The dual reporter strategy expands the time window for graft detection, and combines sensitivity of GNRs with the persistence of a genetic reporter. The findings are novel and validated by the well-established bioluminescence imaging technology.

Essential revisions:

The authors are encouraged to do a subcutaneous injection. This would decouple imaging issues from delivery issues. In addition, from Figure 3A it looks as though most of the injected cells go to the brain. This is confusing – the authors should explain why the cells did not go to the liver or lung and whether they cross the blood-brain barrier.

Minor points:

1) Figure 2A: The genetic map of the reporter constructs lacks a scale bar and does not seem to be drawn to scale. ORFs should be represented such that they are distinguishable from promoters and other genetic elements.

2) Figure 2C: The authors should add a DNA staining such as DAPI, otherwise it is hard to judge cytoplasmic vs. nuclear localization of iRFP720 and luciferase.

3) "Hence, both GNR batches were coated with a silica shell as previously described." Please add reference.

4) The rationale for two types of GNRs is not clear and should be explained.

5) What was the efficiency of transfection with the iRFP construct?

6) "Interestingly, from this stage onwards the bioluminescence images no longer correlated with the photoacoustic imaging (Figure 5A-C)." What do you mean by this point? Time point? If so, please be more explicit.

7) How do detection limits compare between the two techniques? Which is more sensitive? GNRs or iRFP or bioluminescence?

8) Why does the GNR signal drop? Is it because the cells die and release their cargo and macrophages take it to the liver? Or are viable cells taken to the liver.

9) Is there dilution of GNR contrast agent over time as the cells divide?

10) Some additional citations to photoacoustic imaging are appropriate here:a) http://onlinelibrary.wiley.com/doi/10.1002/wnan.1404/fullb) http://downloads.hindawi.com/journals/sci/2016/9240652.pdf

---

## [Author Response]

The authors are encouraged to do a subcutaneous injection. This would decouple imaging issues from delivery issues.

Subcutaneous injections are indeed a good way to validate contrast agents. As far as gold nanorods are concerned, this was done extensively in our previously published article (Comenge et al., 2016). In the case of the genetic reporter, subcutaneous experiments are included (Figure 2—figure supplement 1). To make it clearer to the reader that such experiments have indeed been conducted for both types of labelling strategies, we have inserted the following sentence at the beginning of the section “Imaging biodistribution of cells after intracardiac injection” in Results and Discussion:

“Detection of labelled cells after subcutaneous injection has provided indications that gold nanorods (our previous work^7^) and genetic reporters (Figure 2—figure supplement 1) could enable cell tracking.”

In addition, from Figure 3A it looks as though most of the injected cells go to the brain. This is confusing – the authors should explain why the cells did not go to the liver or lung and whether they cross the blood-brain barrier.

The referee is correct in his/her observation that shortly after intracardiac injection many cells get trapped in the small capillaries of the brain, but also in those of the liver and kidneys as we show in Figure 3A, B and C. In this respect, the initial distribution of cells after injection into the left cardiac ventricle differs from intravenous injection, which, following passage through the right cardiac ventricle, leads to distribution of cells to the lung first, where a majority of cells then become trapped instead. With regard to the question of blood-brain barrier penetration, we show in a different manuscript that i.c. injected mouse MSCs remain inside the lumen of the brain capillaries (please see Figures 2A and D in Scarfe et al., 2017 – this paper has also been uploaded to *eLife* as a ‘Related Manuscript File’). To clarify this aspect, we have now added the following sentence to the revised manuscript:

“As shown in our parallel work (Scarfe et al., 2017, the mMSCs remain in the lumen of the brain capillaries and do not cross the blood-brain barrier.”

Minor points:1) Figure 2A: The genetic map of the reporter constructs lacks a scale bar and does not seem to be drawn to scale. ORFs should be represented such that they are distinguishable from promoters and other genetic elements.

We have not drawn the scheme of the reporter constructs to scale, since some of the elements would then be too small and unrecognisable. However, we have modified the Figure 2A scheme to indicate the reporter gene transcript and the ORFs more clearly. The full sequence of the E2A plasmid version and the positions of all the indicated genetic elements are provided in the Genbank sequence file with accession number MF693179, which is mentioned in the Materials and methods (subsection “Lentivirus vectors”) and Key Resources Table. This plasmid is also available at the Addgene repository (#104587).

We have also modified the legend to Figure 2A, which now reads as follows:

“a) Scheme of the vectors that mediate co-expression of iRFP720 and firefly luciferase either via IRES- or E2A-mediated mechanisms (not drawn to scale). Arrows above and below the scheme indicate the reporter gene transcript and open reading frames, respectively.”

2) Figure 2C: The authors should add a DNA staining such as DAPI, otherwise it is hard to judge cytoplasmic vs. nuclear localization of iRFP720 and luciferase.

We have now repeated the mMSC cellular imaging experiment and included DAPI nuclear staining. The images in Figure 2C have been replaced with a new set of 4 confocal image panels showing DAPI, Luciferase and iRFP720 in greyscale, as well as a merged overlay in colour. Although there is some variability between cells, nuclear accumulation of iRFP720 is clearly recognisable in a number of cells. We have slightly modified the legend for Figure 2C as follows:

“In contrast to Luc, iRFP720 tends to accumulate in the nucleus (yellow triangles).”

Overall, in the context of the in vivo cell tracking studies of this manuscript we would like to note that the subcellular localisation of iRFP720 is less relevant, since it should not affect its optoacoustic detection in animals via MSOT.

3) "Hence, both GNR batches were coated with a silica shell as previously described." Please add reference.

The following sentence with references has been added to the Materials and methods section:

“Silica coating was performed as described in our previous work (Comenge et al., 2016), which was based on slight modifications of a protocol by Fernández-López et al. (Fernández-López et al., 2009).”

4) The rationale for two types of GNRs is not clear and should be explained.

We have added the following sentence in the first section of the Results and Discussion:

“The rationale for evaluating two types of gold nanorods in this study is as follows. […] Second, having two different probes provided us with a powerful control for false positive signals as explained in more detail below in the context of Figure 4. “

5) What was the efficiency of transfection with the iRFP construct?

The transfection efficiency of the cells with the virus particles can vary, depending on the virus purification protocol and their concentration/titre. As mentioned in the Materials and methods (mMSC culture, labelling and immunofluorescence), we selected the iRFP720-positive cells by FACS sorting and expanded these purified populations. We then re-analysed these populations by FACS to compare the iRFP720 signal intensity between the IRES and E2A vector-transfected populations, which is shown in Figure 2B.

To explain this better, we have slightly modified the text in Results as follows:

“Following transfection with lentivirus vectors, the mMSCs were sorted by FACS for iRFP720 fluorescence, to obtain a pure IRES-vector and a pure E2A-vector transfected cell population, respectively, with similar levels of expression of the first ORF protein (iRFP720) (Figure 2B).”

We have also modified the legend of Figure 2 as follows:

“b) FACS analyses of iRFP720 fluorescent signal intensity in purified mMSC populations, which had been transfected with either the IRES or E2A vector, respectively, indicated similar levels of expression of the first ORF.”

6) "Interestingly, from this stage onwards the bioluminescence images no longer correlated with the photoacoustic imaging (Figure 5A-C)." What do you mean by this point? Time point? If so, please be more explicit.

To avoid misunderstandings we have re-worded the sentence as follows:

“Interestingly, after 24 hours, the bioluminescence signal and the photoacoustic signal for GNRs were no longer localised in the same regions/organs (Figure 5A-C), indicating that the vast majority of GNRs followed a different fate than the remaining luciferase-expressing cells.”

Additionally, we modified the legend of Figure 5C as follows:

“(c) In contrast to the kidneys, MSOT signal in liver increased substantially at day 1, although bioluminescence had declined in this tissue at this stage as shown in (b), which suggests that GNRs accumulated in the liver.”

7) How do detection limits compare between the two techniques? Which is more sensitive? GNRs or iRFP or bioluminescence?

We suggest that quantitative determination and comparison of detection limits would be only feasible in highly simplified systems of limited relevance (e.g. phantoms), because factors such as distance to the sensors, differences of ultrasound coupling between different regions of the body, etc. would interfere in the actual measure. However, it is clear from our work and from the literature that the most sensitive technique is bioluminescence, whilst photoacoustic imaging provides much higher resolution. Regarding photoacoustic imaging between GNRs and iRFP, we can also conclude that GNRs generates a much higher contrast, with the disadvantage of potential false positive once cells die. We have slightly edited the conclusion section to make the above points clearer. It now reads:

“We show that multispectral optoacoustic tomography achieves high spatial resolution of the initial cell distribution through analysis of the GNR signal. Whilst expression of iRFP720 is initially undetectable, due to the scattered distribution of cells, it provides long-term tumour monitoring capability in MSOT as clustered iRFP720-positive cells lead to a stronger photoacoustic signal.”

8) Why does the GNR signal drop? Is it because the cells die and release their cargo and macrophages take it to the liver? Or are viable cells taken to the liver.

Our hypothesis is that cells die during the first hours after injection (decrease in total bioluminescence, Figure 5), simultaneously GNR signal moved towards the liver and spleen, which suggest either dead cells, cell debris or cell content are transported there. Since these are phagocytic organs, it is very likely that macrophages are somehow involved, but we lack any definitive proof. So for now, we regard the clarification of this specific aspect as beyond the scope of this work.

9) Is there dilution of GNR contrast agent over time as the cells divide?

Yes. As explained in our previous work (Comenge et al., 2016), this is a relevant limitation of all nanoparticle-based contrast agents. However, we do not observe this phenomenon here due to the massive amount of cell death within the first 24 hours after systemic injection. Reporter genes overcome this potential limitation of nanoparticle-labelling for long term tracking of dividing cells.

10) Some additional citations to photoacoustic imaging are appropriate here:a) http://onlinelibrary.wiley.com/doi/10.1002/wnan.1404/fullb) http://downloads.hindawi.com/journals/sci/2016/9240652.pdf

We were aware of those references, which provide valuable information. We have now added them in the fourth paragraph of the Introduction section (Lemaster and Jokerst, 2017 and Wang et al., 2016).